# Mechanical and Hydration Characteristics of Stabilized Gold Mine Tailings Using a Sustainable Industrial Waste-Based Binder

**DOI:** 10.3390/ma16020634

**Published:** 2023-01-09

**Authors:** Zhenkai Pan, Shaohua Hu, Chao Zhang, Tong Zhou, Guowei Hua, Yuan Li, Xiaolin Lv

**Affiliations:** 1School of Safety Science and Emergency Management, Wuhan University of Technology, Wuhan 430070, China; 2State Key Laboratory of Geotechnical Mechanics and Geotechnical Engineering, Institute of Rock and Soil Mechanics, Chinese Academy of Sciences, Wuhan 430071, China; 3Key Laboratory of Mine Slope Safety Risk Warning and Disaster Prevention and Mitigation, Ministry of Emergency Management, Wuhan 430071, China; 4Wushan Copper Mine, Jiangxi Copper Corporation Limited, Ruichang 332204, China

**Keywords:** gold mine tailings, industrial waste, binder, hydration mechanism, triaxial compressive strength

## Abstract

Sustainable resource utilization of tailings is a long-term challenge. Therefore, a novel waste-based binder is proposed in this study to stabilize/solidify gold mine tailings (GMTs). This binder is composed of fly ash (FA), ground blast furnace slag (GBFS), and metakaolin (MK) activated with mixed calcium carbide residue (CCR) as well as pure reagent grade chemical, sodium hydroxide (SH, NaOH), and plaster gypsum (PG, CaSO_4_·2H_2_O). The mechanical properties and hydration of stabilized tailings with curing period were investigated. Tests included triaxial compression test and nitrogen adsorption to evaluate the strength of the stabilized tailings and microstructure. The results show that the addition of the waste-based binder yields significant improvement in shear strength. Strain softening occurred for all cured samples, and a local shear band can be observed in all failed stabilized samples. Based on the relationship between strength and curing period, it can be speculated that the hydration reaction of the sample ends after around 40 days of curing. A bimodal pore-size distribution was observed in all solidified/stabilized samples. FTIR and ^27^Al MAS-NMR were used to analyze hydration products. The strength improvement of stabilized tailings was mainly attributed to the formation of ettringite and C–S–H gels after various polymerization reactions. These new hydrates bind tailings particles and fill the pores to form a more stable structure, which supplied superior mechanical properties. This paper can provide a theoretical basis for exploring the application of the industrial waste-based binder to modify the mechanical properties of gold tailings.

## 1. Introduction

Tailings are the residual wastes formed by mineral processing of ores. They are stored mainly in situ by building artificial tailings ponds [1,2]. With the increasing demand for sand in engineering as well as the improvement of engineering cost, using tailings instead of river sand in engineering applications presents a great prospect. When tailings are not properly treated, their long-term deposition can lead to water and soil pollution [3,4]. Land-use restriction policies will demand the raising of tailings dams’ height, which will require more management and operational costs. Therefore, increasing the use of tailings is a favorable research direction for the economy. In order to ensure the economic, effective, and environmentally friendly treatment of tailings, two major research works are being pursued. On the one hand, tailings can be used as engineering materials. Li et al. [5] proposed an environmentally friendly 3D printing building material made of copper and iron tailings, which matched the extrusion printing process. Young and Yang [6] studied the application of iron tailings powder-sintered cement clinker, indicating that iron tailings powder has good biological activity. Cement clinker produced with tailings as raw material has been proven to have good physical properties and shows great development potential. Researchers have improved mining and beneficiation processes to make the ores more user-friendly [7,8]. This leads to finer granularity of tailings produced but poor consolidation and permeability after tailings deposition [9]. Irregular settlement of tailings in tailings ponds causes significant differences in compaction, which may result in high saturation and great compressibility [10]. Then, it has the weakness engineering characteristics of large deformability and poor strength, and it poses the risk of heavy metal ions leaching out. Further, it is not suitable for direct application as engineering materials. On the other hand, tailings can be used as an additive for traditional engineering materials, for example, as aggregate for concrete [11] and as partial sandy soil for cement mortar [12].

In recent years, engineering materials have been modified by adding curing agent or binder, which has attracted the attention of many researchers. At present, the curing materials used in soil are mainly divided into inorganic binder, ionic soil stabilizer, and compound curing agent. Inorganic binders mainly include cement, lime, fly ash, fiber, etc. The ionic soil stabilizer is a chemical substance with strong ionization. It interacts with the soil and removes the adsorbed water in the clay minerals, changing the soil from hydrophilic to hydrophobic and effectively improving its engineering properties. However, these curing agents have shortcomings in sustainability and durability. For example, the production of cement consumes a great amount of energy and produces a large amount of carbon dioxide, which is not in line with the concept of energy conservation and emission reduction [13]. Fiber as a curing agent has a high cost, and its durability, heat resistance, and impact resistance are relatively poor [14]. The ionic soil stabilizer still has some disadvantages, such as low strength and low compatibility as a direct subgrade base and being water repellent [15]. The preparation of compound curing agent is usually complicated, with high technical requirements.

As an alternative, geopolymers have a wide range of sources and sustainability as binders and have been used by researchers to stabilize soils for experimental studies. Currently, geopolymers are mostly one-part or two-part compositions. Çevik et al. [16] investigated the durability and mechanical performance of fly ash-based geopolymers with nano silica subjected to sulfuric acid, magnesium sulfate, and seawater solutions and found that nano silica fly-ash-based geopolymer had superior durability compared to OPC under chemical attacks. Du et al. [17] explored the physical, hydraulic, and mechanical properties of clay stabilized by lightweight slag-soil geopolymer and showed that the engineering performance of geopolymer-stabilized soil was superior to that of cement-stabilized soil in terms of water absorption, permeability, and strength. Kurtoglu et al. [18] studied the mechanical and durability properties of fly-ash- and slag-based geopolymer as concretes and found that slag-based geopolymer concretes were stronger and more durable than fly-ash-based geopolymers due to their more stable polymerization structure. Composite geopolymers can be produced by mixtures of natural materials such as clinker, clay minerals, or gypsum, or they can be obtained from industrial waste residues, including bag dust, fly ash, industrial alkali, blast furnace slag powder, calcium carbide residue, scrubber sludge, and red mud. However, there have been few studies on the effects of composite geopolymer treatment on stabilized gold mine tailings. Composite geopolymers can effectively improve the mechanical properties of such tailings materials. They use materials with pozzolanic properties to activate clay minerals into hydrates through activators (alkalis) [19,20], which is essentially a polymerization reaction. Therefore, it is imperative to study the solidification/stabilization of gold mine tailings with binders composed of waste-based composite geopolymers.

Taking into account the economic limits of waste-based binder and the likelihood of having it accepted for mining enterprises, a content ratio of 5% binder and 95% gold tailings was used to prepare samples for experimentally investigating the mechanical properties and hydration mechanism of gold mine tailings modified by this sustainable waste-based binder. Mechanical tests and phase-structure analyses were carried out on the hydrated samples after different curing periods. Triaxial compression tests were conducted to study its mechanical behavior. The microstructure of the hydrated samples was characterized by nitrogen-adsorption tests. Fourier-transform infrared spectroscopy (FTIR) and ^27^Al magic angle spinning nuclear magnetic resonance (MAS-NMR) were used to analyze the hydration products of blending gold mine tailings with binder.

## 2. Materials

### 2.1. Industrial Wastes as a Binder Preparation

The industrial waste-based binder developed in this study is composed of fly ash (FA), ground blast furnace slag (GBFS), calcium carbide slag (CCR), metakaolin (MK), sodium hydroxide (SH, NaOH), and plaster gypsum (PG, CaSO_4_·2H_2_O) mixed in a certain proportion. The composition matrix material of the binder is shown in Figure 1.

The formula ratio of chemicals used for industrial waste-based binder production is presented in Table 1. The FA was obtained from a local power plant, of which the material with a particle size of less than 45 μm accounted for 80%. The GBFS was collected from a steel plant and then air-dried at 60 °C for 24 h. The MK was formed by dehydrating kaolin at high temperatures (600~900 °C) to break the van der Waals bonds between the layered silicate structures. The CCR was acquired by drying an industrial waste liquid from a gas refinery at 105 °C for 8 h. The PG, a white powder, was purchased from a by-product company in Hubei Province, China. The SH, consisting of superior-grade pure material with glass bead-like solid particles, was purchased from a chemical products company in Cangzhou City, China. GBFS was mechanically ground in a ball mill (Ransbach-Baumbach, Stuttgart, Germany) with a grinding medium of 40 stainless-steel balls to increase its specific surface area (SSA). Overall, the pozzolanic activity of the GBFS increases with the increase of SSA. However, the grinding consumes time and money. The influence of SSA on curing efficiency was explored through preliminary tests. The strength of stabilized tailings does not increase significantly when the SSA is higher than 3.5 m^2^/g. Therefore, in this study, it was selected as 3.5 m^2^/g, and its activity index is S95. The physicochemical properties of matrix material of the binder are summarized in Table 2.

### 2.2. Gold Mine Tailings

Tailings collected from a tailings dam in Longnan City, China, were used in this study. The samples collected were dried at 105 °C. During the operation of the tailings reservoir, the granularity and density of tailings changed with the length of the dry beach. To check repeatability, raw tailings materials obtained at five different positions along the dry beach surface were tested. The engineering and environmental characteristics of the tailings were determined. The average of the test results is summarized in Table 3. X-ray fluorescence (XRF) spectrometer (xSORT, SPECTRO, Kleve, Germany) was used to analyze the chemical composition of the gold mine tailings (GMTs) and binder components. The main chemical composition is summarized in Table 4. An X-ray diffractometer (XRD; Bruker AXS-D8, Saarbrucken, Germany) with 40 kV and 135 mA (Co Kα radiation) was used to determine the mineralogy of the tailings. The mineralogy of gold tailings was analyzed by quantitative Rietveld XRD method and was identified by comparing the diffraction patterns data of PDF2004 card in Jade software (Jade 6.0, Materials Data, CA, USA). The raw gold tailings are mainly composed of quartz and illite, with a content exceeding 80%. Other mineral components are clinochlore, calcite, pyrite, and galena in descending order of content, as shown in Figure 2.

## 3. Sample Preparation

The raw composition material of the binder was weighed according to the designed formula and mixed in an electric mixer for 20 min, and then, 5% content binder was dry-mixed with gold tailings for 30 min to achieve the homogeneity of the sample. The optimal water content (*w*_opt_) and maximum dry density (ρ_max_) of the blends of binder with tailings were obtained by the Proctor standard compaction test. Preliminary test results showed that the *w*_opt_ and ρ_max_ of the admixed tailings with 5% of binder dosage were 17.8%, and 1.72 g/cm^3^, respectively. The water content (or binder dosage) is defined as the ratio of the weight of water (or binder) to that of the dried gold tailings according to the designed ratio of cementing agents and soil [21]. Cylindrical samples with 39.1 mm diameter and 80 mm height were prepared for triaxial compression tests. Deionized water with a pH of 6 was added to the dry blends according to their optimal water content. Then, the mixtures containing the tailings, binder, and water were thoroughly agitated in an electric mixer to achieve uniformity. Pre-compaction was conducted to obtain the initial compression state, with the initial dry density of the sample set to 1.5 g/cm^3^.

The authors have described the procedures for preparing solidified/stabilized samples in Pan et al. [22]. A cylindrical sample was extruded from the stainless-steel mold, sealed in a polythene bag, and placed in a standard curing box (temperature 25 ± 2 °C, relative humidity 90%). The solidified/stabilized samples were cured for 3, 7, 14, 28, and 90 days (d) before carrying out the triaxial compression, nitrogen adsorption, FTIR, and ^27^Al MAS-NMR tests. For each curing period, four identical independent samples were prepared (Figure 3). One of these was used to measure dry density and moisture content during curing as well as to study the microstructure and hydration mechanism, while the other three were used to evaluate the variation in mechanical strength over curing time.

## 4. Apparatus and Test Procedure

The triaxial compression strength (TCS) test was carried out on a strain-controlled triaxial apparatus. Samples were immediately sheared under confining pressures of 100, 200, and 400 kPa after each curing periods with a constant axial strain rate of 0.06% /min as per ASTM D2850-15 [23]. The unconsolidated undrained shear test was carried out on the stabilized/solidified samples.

After completion of the tests, the failed stabilized/solidified samples were ground into powder for FTIR and ^27^Al MAS-NMR analysis to identify the hydration products.

A Fourier-transform infrared spectrometer (FTIR; Thermo Nicolet Corporation, Nicolet Nexus 410, Madison, Wisconsin, USA) was used to record infrared spectra of the hydrated samples using the KBr pellet technique. The wavenumber range is 4000–400 cm^−1^, with a 2 cm^−1^ spectrometer resolution and a letter-to-column ratio of 5000. ^27^Al solid-state MAS-NMR tests were conducted on the stabilized/solidified samples using a Bruker Avance III 400 MHz spectrometer (Bruker, Saarbrucken, Germany), operating at Larmor frequency of 104.3 MHz and magnetic field of 9.4 T. A 4 mm O.D. zirconia rotor chemical magnetic probe was used for magic angle spinning. NMR experiments were performed with a 1 µs pulse width, a 1s pulse delay, and a spinning speed of approximately 12 kHz and 2048 scans.

The broken samples with similar mass in triplicate were immersed in acetone to prevent further hydration and then freeze-dried in a vacuum oven for 24 h. Nitrogen-adsorption tests were carried out on the selected samples to characterize microstructure using a pore-size analyzer (Quantachrome, NOVA 1000e, Beijing, China). The tested samples were shaped into 2 × 2 × 2 mm blocks for nitrogen-adsorption tests. The results were expressed as the cumulative pore volume and the differential between the pore volume and the logarithm of pore size using the Barrett–Joyner–Halenda (BJH) method.

## 5. Results and Analysis

### 5.1. Mechanical Properties of Gold Tailings Blended with Binder

#### 5.1.1. Stress-Strain and Deformation Characteristics

Figure 4 shows the stress-strain curves and failure deformation of gold tailings stabilized/solidified by waste-based binder after different curing times. The peak deviatoric stress and initial elastic modulus increased with curing time regardless of the confining pressures. For the raw gold tailings sample, the stress–strain relationship changed from strain softening to strain hardening with the increase of confining pressure. These results are consistent with previous findings [24].

For all the cured specimens, strain softening occurred to varying degrees. The deviatoric stress increased rapidly with the axial strain during the initial small deformation and then reduced beyond the peak point. When it reached a large axial strain, the deviatoric stress remained at a plateau, in which the sample was in the quasi-steady state (critical state) [25]. Only the samples cured for 3 d under confining pressure of 200 kPa and the samples cured for 3 and 7 d under a confining pressure of 400 kPa did not reach this quasi-steady state at the end of shearing, and their post-peak deviatoric stress gradually decreased.

An X-type local shear zone [26] can be observed in the hydrated gold tailings sample under the confining pressure of 100 kPa. Under the confining pressure of 200 kPa, the failure of the specimen still exhibited a transverse development but, overall, also presented a cross shear band. Their failure modes were bulging deformation. Under the confining pressure of 400 kPa, a single local shear band can be seen in the sample, accompanied by shear slip failure [27]. The shear strength of the raw tailings specimen was close to the residual strength of the early-cured specimens, but the difference between them increased with the increase of curing time and confining pressure. This is because although the early hydration reaction enhanced the shear strength, the overall structure of the hydrated tailings sample did not change significantly. The formation of new hydration products changed the distribution of particles and pores and enhanced the residual strength with increased curing time.

#### 5.1.2. Strength Characteristics

Figure 5 illustrates the variations of the TCS of the stabilized gold tailings with curing time. It can be seen that the TCS of samples has obvious anisotropy induced by hydration reaction. The TCS of all samples increased rapidly within 28 d of curing. During the curing period from 28 to 90 d, the TCS of the samples under all confining pressures did not increase significantly. It is assumed that the strength of the stabilized samples was the highest at the end of hydration. By data analysis of the nonlinear curve fitting in OriginLab 2018 software, a relationship model between the TCS and curing time can be established by Gaussian fitting [28] as follows (Equation (1)):(1)$$TCS=s+a*P_{0}*\exp(-0.5*{\left((x-x_{c})/w\right)}^{2})$$
where *x* is the curing time, *P*_0_ is the atmospheric pressure, and *s*, *a*, *x_c_*, and *w* are fitting parameters.

Table 5 lists the fitting parameters changes of *s*, *a*, *x_c_*, and *w* under different confining pressures. It can be seen from Figure 4 that there is a good consistency between the test data and the Gaussian fitting curves. The difference of standard deviation is tiny under all confining pressures. The model can well predict the variation of shear strength with curing time. Parameter *s* is the maximum shear strength within 90 days of curing. Parameter *x_c_* increases with the increase of confining pressure, and *w* decreases with the increase of confining pressure. The variation of parameter *a* has little correlation with the confining pressure. After curing for 42 d, the enhancement of TCS is no longer obvious. It can be inferred that this is the termination time of the hydration reaction.

Figure 6 shows the variation of dry density and water content of waste-based binder solidified/stabilized gold tailings with curing time, which trend opposite to each other. According to the Gaussian model, the change of dry density and water content from the end of the hydration to 90 d of curing period is caused by the evaporation of water in the solidified/stabilized sample. They both vary insignificantly during the last 50 d of curing due to maintenance of the environment at 90% air relative humidity. Figure 7 illustrates the variations of the TCS of solidified/stabilized samples with dry density under all confining pressures. There is a positive correlation between the TCS and dry density of the samples, indicating that the hydration reaction produces denser products, leading to an increase in strength. Combined with Figure 6, the increase in dry density caused by water evaporation after hydration does not significantly enhance the TCS. It can be concluded that the new hydrates have higher density and hardness, which makes the overall structure of the sample more stable. Evaporation of water does not change its true fabric. Furthermore, the TCS first increases slowly and then increases rapidly with the dry density increasing, with the dry density of 1.493 g/cm^3^ being recognized as a threshold (critical value) between slow and rapid increase of strength.

#### 5.1.3. Strength and Deformation Parameters

Table 5 indicates the variations of the strength and deformation parameters of solidified/stabilized gold tailings. The cohesive strength and internal friction angle are calculated using the Mohr–Coulomb strength criterion [29], as shown in Equation (2).
(2)$${\left(\sigma_{1}-\sigma_{3}\right)}_{f}=\frac{2}{1-\sin\varphi }(c\cos\varphi +\sigma_{3}\sin\varphi )$$
where ${\left(\sigma_{1}-\sigma_{3}\right)}_{f}$ is shear strength, c is the cohesive strength, φ is the internal friction angle, and $\sigma_{3}$ is the confining pressure.

It is seen from Table 6 that the cohesive strength increases with the extension of curing period, which is consistent with the result that the strength increases further after 28 d of curing. The internal friction angle increases with the increase of curing time after 3 d of curing. This explains why the hydration reaction changed the structure between tailings particles. Although more cohesive strength is provided in the early stage of hydration, the hydration also makes the particles smoother, weakening their interlocking and sliding friction. The secant modulus and failure strain are important parameters for deformation analysis that have a significant relationship with shear strength. The secant modulus increases with the increase of curing time, and those of all samples are within the range of 30–140 MPa.

It is worth noting that, for the solidified/stabilized samples under confining pressure of 200 kPa, the secant modulus of samples is larger than that of samples under confining pressures of 100 kPa and 400 kPa. In general, the modulus of elasticity of the solidified/stabilized sample increases with the increase of confining pressure. However, when the confining pressure exceeds a certain value, due to the small initial density and the insufficient compactness of the sample, when the deviatoric stress on the solidified/stabilized sample further increases to reach the TCS, its plastic deformation increases. The proportion of plastic strain in the breaking strain of the solidified/stabilized sample is higher than that of elastic strain, so the secant modulus does not increase with the increase of confining pressure. The breaking strain decreases with the increase of curing time and increases with the increase of confining pressure in almost all samples. This is inconsistent with the brittle failure of hard rock, in which the greater the confining pressure, the smaller the failure strain. The reason is that the increase of confining pressure enhances the ductility of the samples, and the growth of deviatoric stress slows down before reaching the peak strength; that is, the plastic strain increases, and an obvious yield platform [30] appears (Figure 4c). The essence of plastic deformation is the relative slip or shear of the material. It can be seen from the failure mode of the solidified/stabilized sample. This phenomenon is also found when other cement-based cementitious materials enhance the mechanical properties of soft soil [31,32].

### 5.2. Microstructural Analysis

#### 5.2.1. Particle Characteristics of Solidified/Stabilized Gold Tailings

Nitrogen adsorption is a common test method to study the microscopic pore structure of materials. When nitrogen molecules reach adsorption equilibrium on the solid surface, the specific surface area of the particles can be calculated by Equation (3):(3)$$S_{g}=\frac{N\delta V_{m}}{22400w}$$
where *N* is Avogadro’s constant, the number of molecules in a substance per unit of mass, is 6.024 × 10^23^ mol^−1^; δ is the cross sectional area of the nitrogen molecule, nm^2^; $V_{m}$ is the adsorbed volume of a monolayer nitrogen molecule on the inner pore surface of the solidified/stabilized sample, cm^3^; w is the mass of the sample tested, g.

It is worth noting that monolayer nitrogen molecules are not adsorbed on the pore surface of the material. It is assumed that the adsorption heat of nitrogen molecules in the first layer of the inner pore surface is a constant and that of the other layers is different. The BET (Brunauer, Emmett, and Teller) equation (Equation (4)) can be used to calculate the true nitrogen volume in the pores of the material on the basis of thermodynamic and kinetic analysis.
(4)$$\frac{P}{V(P_{0}-P)}=\frac{1}{V_{m}C}+\frac{C-1}{V_{m}C}\left(\frac{P}{P_{0}}\right)$$
where *V* is the nitrogen volume adsorbed in the pores of a 1 g sample, cm^3^; *P* is the pressure of nitrogen injection, MPa; $P_{0}$ is the saturation vapor pressure at the boiling point of nitrogen, MPa; C is the adsorption thermal constant, and the adsorption capacity is stronger when its value is larger; the range of $P/P_{0}$ is 0.05 to 0.35.

Figure 8 presents the BET curves of waste-based binder solidified/stabilized gold tailings. The $V_{m}$ and *C* can be calculated from the slope and intercept of the BET curves. Combined with Equation (3), the specific surface area of solidified/stabilized gold tailings can be obtained. The specific surface area of the samples without curing and with curing for 3, 7, 14, 28, and 90 d are 3.03, 5.58, 6.67, 8.21, 8.36, and 6.93 m^2^/g, respectively. This indicates that the solidified/stabilized gold tailings are dissolved into smaller fineness particles by the waste-based binder, which is consistent with the expectation that the higher the specific surface area of the particles in the appropriate range, the higher the strength will be [33]. However, the specific surface area of the solidified/stabilized samples cured for 90 d decreases. They may have been oxidized by further contact with the air after the hydration reaction, resulting in smaller specific surface area and instability of the structure. The adsorption heat refers to the thermal effect produced by adsorption [34]. Specifically, when the molecules move to the solid surface, their movement speed will be greatly reduced, thus releasing heat in the process of adsorption. The adsorption thermal constants of the samples without curing and curing for 3, 7, 14, 28, and 90 d are 336.99, 207.28, 214.55, 164.73, 142.01, and 64.44, respectively. The results show that the adsorption capacity of the particle surface of solidified/stabilized materials becomes weaker with the increase of curing time. This is related to the type and amount of hydration products.

#### 5.2.2. Pore-Size Distribution

Figure 9 shows the changes of the pore-size distribution in the waste-based binder solidified/stabilized gold tailings. The ordinate is plotted as V*_log-differential_* (V*_log-differential_* = dV/d(*logr_i_*)), where V is the pore volume (mL/g) of dry solidified/stabilized gold tailings per unit mass, and *r_i_* corresponds to the pore diameter (nm). A bimodal pore-size distribution can be observed in all solidified/stabilized samples except for uncured samples. The two dominant pores in the solidified/stabilized samples are defined as intra-aggregate pores and interaggregate pores. With the extension of curing time, the proportion of intra-aggregate pores reduces, and the pore size of the dominant pores moves to the left. The pore diameter of dominant interaggregate pores presents trivial changes when the curing period extends from 3 to 90 d. This indicates that the early hydration causes agglomeration in the samples, and they become more compact with an increase of curing duration.

### 5.3. FTIR Analysis

FTIR spectra of solidified/stabilized gold tailings hydrated for different durations are displayed in Figure 10. Different absorption bands in the spectra presenting analogous IR spectra are seen at different hydrated times. Except for the unhydrated samples, the absorption bands of all cured samples between about 3618 and 3627 cm^−1^ are related to the Ca-OH stretching vibration in calcium hydroxide (Ca(OH)_2_) [35]. This absorption band decreased from 3 to 90 d of curing, indicating that the content of Ca(OH)_2_ during hydration decreased with the curing time. This is consistent with the expected reduction in alkali content. The absorption band at near 1432 cm^−1^ of the raw gold tailings sample is related to the asymmetric stretching vibration of carbonate (CO_3_^2−^) [36], reflecting the XRD analysis results that the raw tailings contain calcite. Since carbonization of hydrated samples can be avoided by placing them in sealed plastic bags, the absorption band of CO_3_^2−^ ions corresponding to all solidified/stabilized samples shifts to the high wavenumber direction, moving to approximately 1437 cm^−1^. The results reveal that calcite does not participate in the hydration process, and the electronegativity of adjacent groups is enhanced, and the molecular chains are shortened. The absorption band at 995 cm^−1^ is related to the asymmetric Si–O stretching vibration of the SiO_4_ tetrahedron. The absorption band center of the solidified/stabilized sample gradually shifted towards the lower wave number of 977 cm^−1^ after 90 d of curing that forms new hydration products, suggesting the presence of a framework of aluminosilicate glasses (C–S–H gel). Generally, for the C–S–H gel formed by the hydration of ordinary Portland cement (OPC), there is a characteristic band in the frequency of 960–970 cm^−1^ [37], whereas the absorption peak of C–S–H gel here produced in the solidified/stabilized gold tailings is at a higher frequency of 977 cm^−1^, representing a more complex polymeric structure.

In the FTIR spectra, the absorption peak at 776 cm^−1^ is associated symmetric Si–O stretching vibration within quartz. Although quartz does not participate in hydration reactions, its inter-molecular hydrogen bond formation makes the stretching vibration. The formation of C–S–H gel and hydrogen bond are the key to improve the mechanical strength. In the infrared spectra of the sample cured for 3, 7, and 14 d, the unnoticeable band at 522 cm^−1^ is attributed to the O–Si–O out-of-plane bending vibration in the SiO_4_ tetrahedron of calcium silicate (Ca_2_SiO_4_). However, this absorption band was not detected in the infrared spectra of the samples cured for 28 and 90 d, indicating that Ca_2_SiO_4_ was hydrated into the polymeric structure [38]. This process shows that the hydration products gradually form micro-aggregate and then slowly fill the pores and bond the unreacted tailings particles, making the stabilized sample denser, which can be verified in terms of its mechanical strength [39,40].

The broad absorption bands between 420 and 463 cm^−1^ are related to the Si-O in-plane bending vibration of the SiO_4_ tetrahedron within illite [41]. The frequencies of absorption bands decrease with the increase of curing time, which is caused by the induction effect of more electron-donating groups released by illite after hydration. In addition, it is worth noting that there is no band of 1622 cm^−1^ corresponding to H-O-H vibration of interlayer water and no band of 3407 cm^−1^ corresponding to OH^−^ ions vibration in structural water because the sample is freeze-derived.

### 5.4. ^27^Al MAS-NMR Analysis

^27^Al MAS-NMR technology can be used to analyze the coordination state of aluminum in the crystal structure of solidified/stabilized samples to determine the type of aluminate [42]. Figure 11 shows ^27^Al MAS-NMR spectra of solidified/stabilized gold tailings cured for 3 and 90 d, respectively. According to previous studies [43], the chemical shifts of 80~50 ppm, 30~25 ppm, and 15~0 ppm resonances represent tetrahedral-, pentahedral-, and hexahedral-coordinated aluminum, respectively. The width of resonance at these chemical shifts reflects the amorphous characteristics of the waste-based binder solidified/stabilized gold tailings sample. The resonances within 75~60 ppm are associated with tetrahedral-coordinated aluminum included in the C_3_A, alite, and C–S–H.

The aluminum that substitutes the tetrahedral sites of silicate chain in C–S–H gel can produce a band of approximately 74 ppm [44]. Combined with the spectrum, it can be seen that C–S–H gel includes calcium silicate hydrate and calcium aluminum silicate hydrate. The fivefold aluminum is present between the interlayer of C–S–H, and it shows a band centered at 34 ppm. No band of five-coordinated aluminum is found in the hydrated samples. The peak at 10.2 ppm corresponds to the hexahedral-coordinated aluminum in AFt (ettringite), and the peak of the hexahedral-coordinated aluminum in AFm is located at −5.2 ppm. After 90 d of curing, there is a weaker peak of hexahedral-coordinated aluminum in AFm than that in the sample cured for only 3 d. It is possible that the hydroxyl groups of the waste-based binder are sufficiently reacted with monosulfate-rich AFm to form AFt. Moreover, compared with the samples that were cured for only 3 d, the peaks at around 60~75 ppm and 10~15 ppm of the samples cured for 90 d were relatively enhanced, and the range of these peaks became wider. The spectrum of the cured samples also includes different residual coordination aluminum peaks of 160~150 ppm and −70~−60 ppm in unreacted illite.

## 6. Conclusions

This study aims to systematically explore the sustainable utilization of an industrial waste-based binder, which consists of fly ash (FA), ground blast furnace slag (GBFS), and metakaolin (MK) supplemented by calcium carbide residue (CCR) and sodium hydroxide (SH, NaOH) activation and coupled with plaster gypsum (PG, CaSO_4_·2H_2_O) and CCR to provide a calcium source in order to form a geopolymer with higher strength. It is used to solidify/stabilize tailings from mine-beneficiation activities to improve their properties. The main conclusions can be drawn according to analysis results as follows:All the solidified/stabilized gold tailings show strain softening. A local shear band is observed in these samples due to the change of particles and pores distribution. The curing period of 42 d is considered the end time of the hydration reaction. The strength parameters and secant modulus of all solidified/stabilized samples increase with increasing curing period, whereas the breaking strain decreases with it.The micro-structural analysis of waste-based binder solidified/stabilized gold tailings helps explain the effect of hydration on the microscopic mechanism of strength enhancement. The adsorption capacity of the particle surface of solidified/stabilized materials weakens with the increase of curing time. A bimodal pore-size distribution characteristic was found in solidified/stabilized gold tailings samples.The hydration products formed at room temperature are grossly C–S–H gel and AFt (ettringite). They facilitate the formation of micro-aggregates, thus providing better mechanical strength to the solidified/stabilized sample. From ^27^Al MAS-NMR analysis results, the formation of tetrahedral-coordinated aluminum in C–S–H gel and hexahedral-coordinated aluminum in AFm could be observed in hydrated samples. This novel industrial waste-based binder is proven to be economical, reliable, and environmentally friendly and can broaden the applications of tailings in engineering.

## Figures and Tables

**Figure 1 materials-16-00634-f001:**
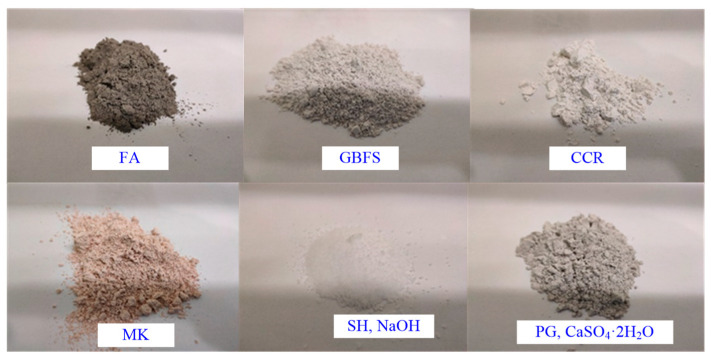
Raw material components of the waste-based binder. FA, fly ash; GBFS, ground blast furnace slag; CCR, calcium carbide slag; MK, metakaolin; SH, sodium hydroxide; PG, plaster gypsum.

**Figure 2 materials-16-00634-f002:**
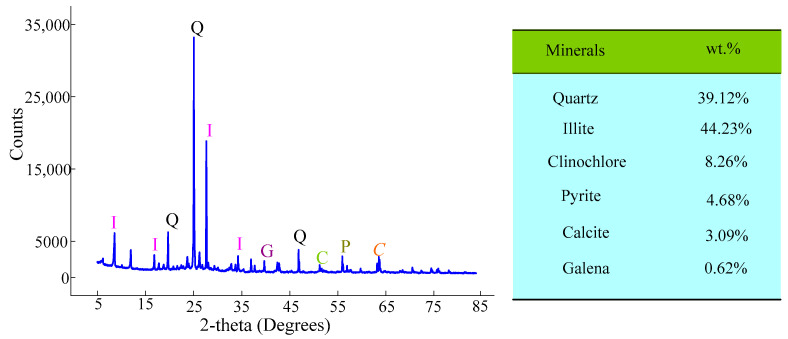
XRD pattern and mineralogical compositions of gold mine tailings.

**Figure 3 materials-16-00634-f003:**
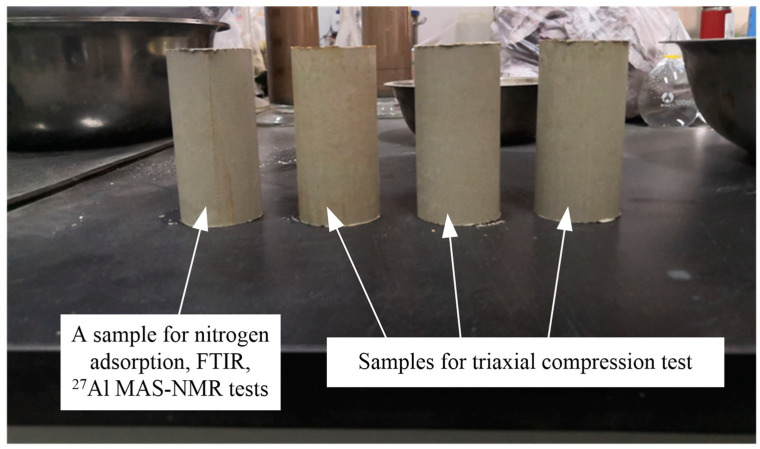
Four identical samples in laboratory (samples cured for 14 days case).

**Figure 4 materials-16-00634-f004:**
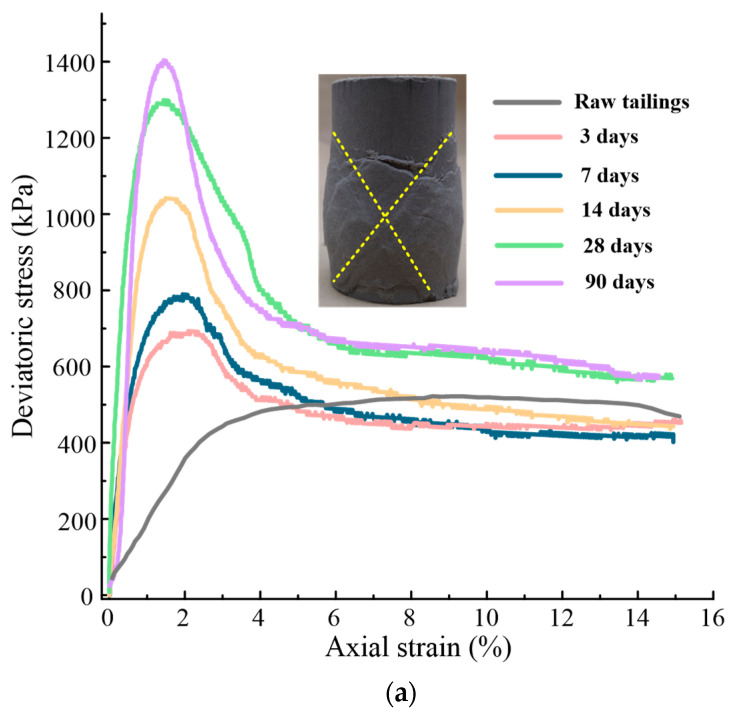
Stress-strain curves and failure deformation of the stabilized/solidified gold tailings: (**a**) under confining pressure of 100 kPa, (**b**) under confining pressure of 200 kPa, and (**c**) under confining pressure of 400 kPa.

**Figure 5 materials-16-00634-f005:**
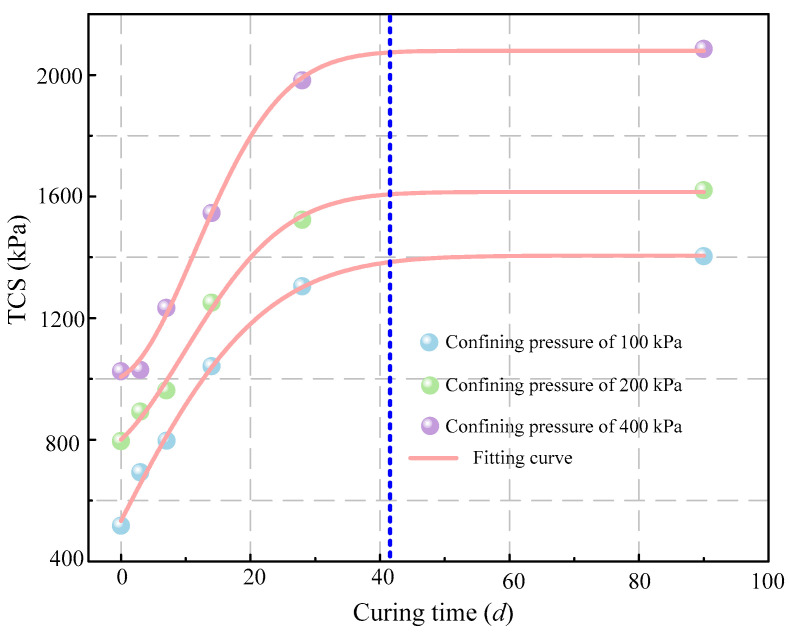
TCS (triaxial compressive strength) of hydrated sample after different curing times under three different confining pressures.

**Figure 6 materials-16-00634-f006:**
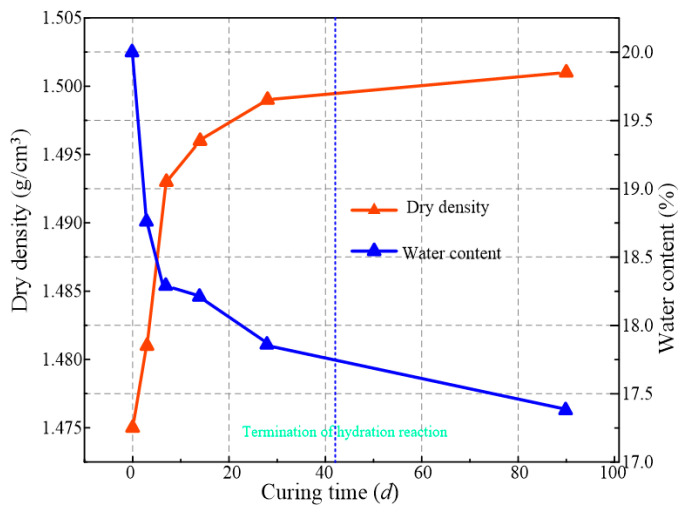
Variation of dry density and water content of waste-based binder solidified/stabilized gold tailings.

**Figure 7 materials-16-00634-f007:**
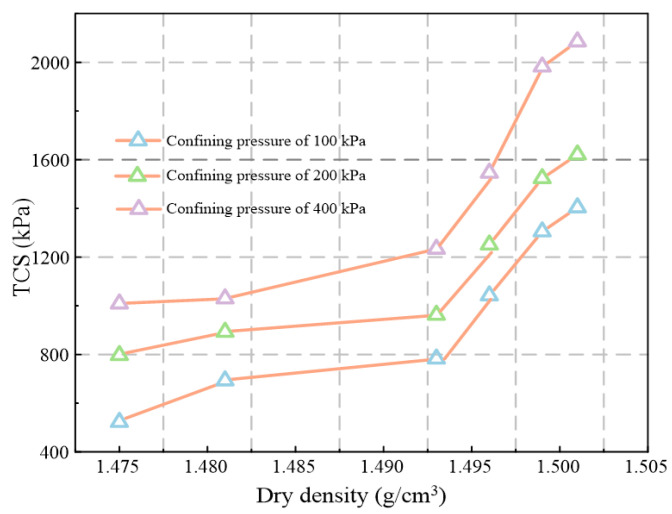
Relationship between TCS and dry density at three confining pressures.

**Figure 8 materials-16-00634-f008:**
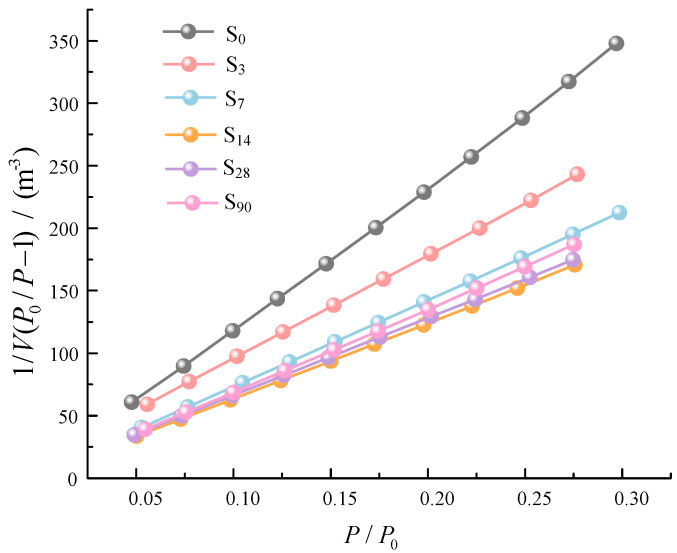
BET (Brunauer−Emmett−Teller) curves of the solidified/stabilized gold tailings after different six curing periods (0 to 90 d).

**Figure 9 materials-16-00634-f009:**
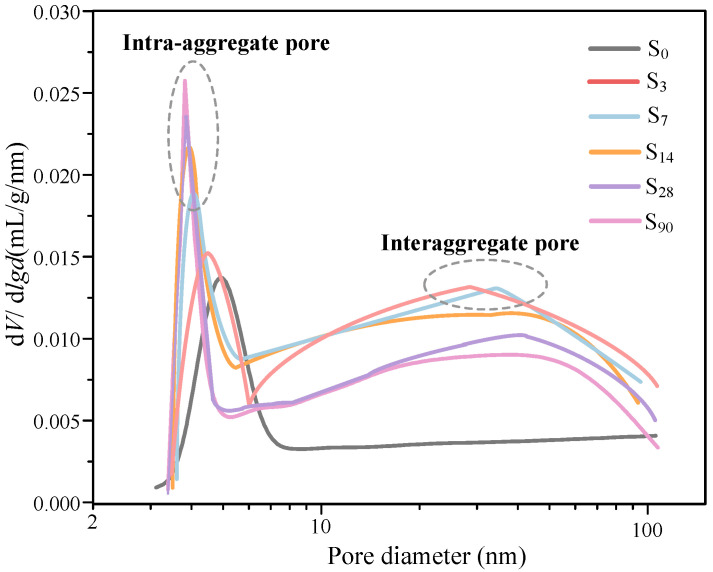
Pore-size distribution curves of solidified/stabilized gold tailings after six different curing times.

**Figure 10 materials-16-00634-f010:**
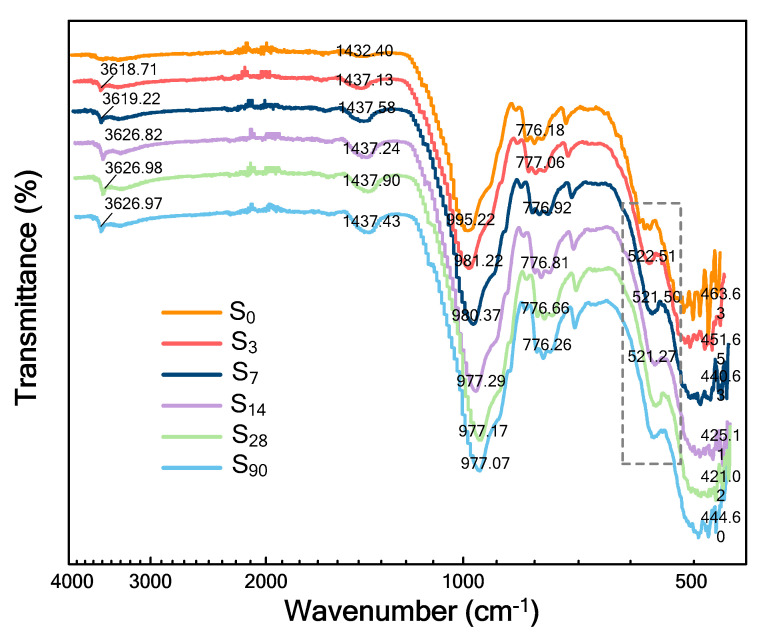
FTIR spectra of waste−based binder solidified/stabilized gold tailings hydrated for six different durations (0 to 90 d).

**Figure 11 materials-16-00634-f011:**
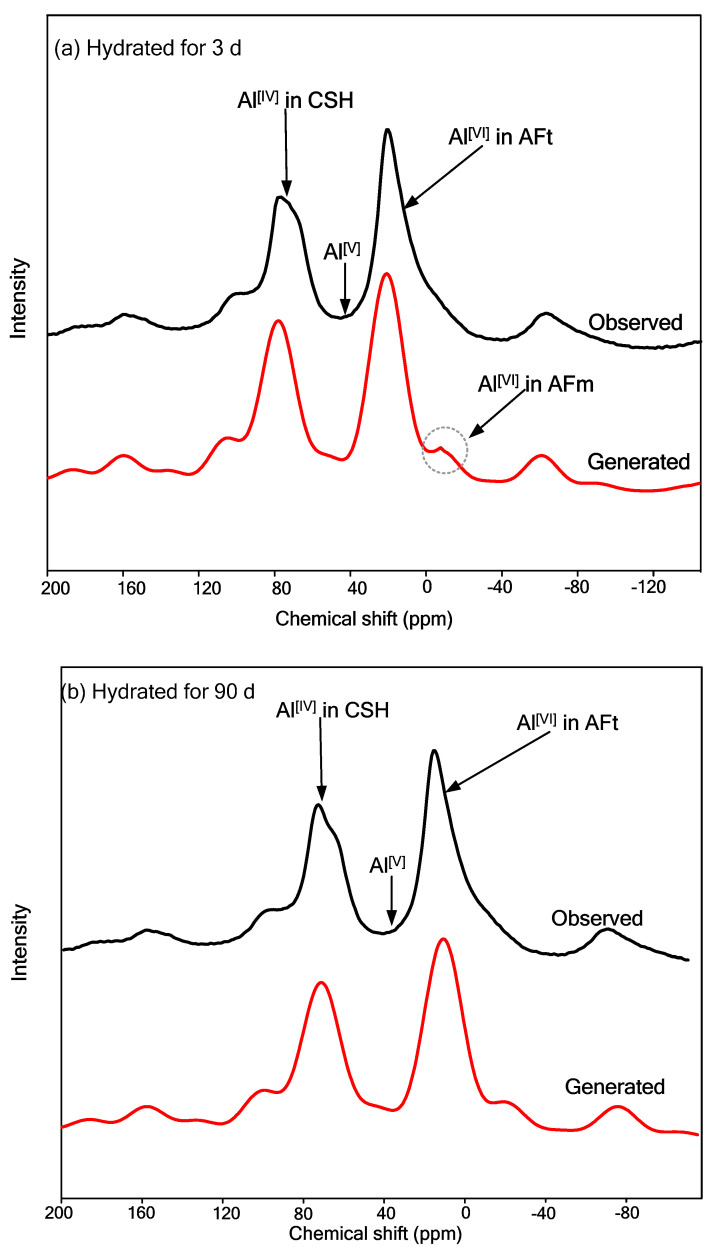
^27^Al MAS−NMR spectra of solidified/stabilized gold tailings: (**a**) cured for 3 d; (**b**) cured for 90 d.

**Table 1 materials-16-00634-t001:** Content of chemicals used for industrial waste-based binder production.

Chemicals	FA	GBFS	CCR	MK	SH	PG
Content (%)	10.53	57.89	7.37	15.79	3.16	5.26

**Table 2 materials-16-00634-t002:** Basic physicochemical properties of matrix material of waste-based binder.

Test Parameters	Material						Test Standards
	FA	GBFS	CCR	MK	SH	PG	
Specific gravity, Gs	2.71	3.13	2.11	2.39	2.22	2.54	ASTM D854-14
Specific surface areas (m^2^/g)	4.13	3.58	2.32	10.23	1.51	8.24	China GB/T 8074-2008
pH	10.34	11.06	12.37	6.38	14.0	7.0	ASTM D4972-18
Particle size distribution (%)							ASTM D422-63
<0.005 mm	12.36	13.65	10.21	39.75	0	20.36	
0.005–0.075 mm	80.05	69.46	51.92	60.25	20.13	79.64	
0.075–0.5 mm	7.59	16.89	37.87	0	79.87	0	

**Table 3 materials-16-00634-t003:** Basic physicochemical properties of raw tailings material.

Test Parameters	Value	Test Standards
Specific gravity, Gs	2.78	ASTM D854-14
Maximum dry density (g/cm^3^)	1.67	ASTM D698-12
Minimum dry density (g/cm^3^)	1.31	
Optimum water content (%)	19.7	
Liquid limit, *w*_L_ (%)	39.23	ASTM D4318-10
Plasticity limit, *w*_P_ (%)	20.71	
Initial water content (%)	15.9	ASTM D2216-19
Soil pH	8.56	ASTM D4972-18
Soil classification	CL	ASTM D2487-17
Particle size distribution (%)		
Sand particle (>75 µm)	1.2	ASTM D422-63 ASTM D2487-17
Silt particle (5–75 µm)	78.2	
Clay particle (<5 µm)	20.6	
Particle size *d*_50_ (µm)	15	
Uniformity coefficient, *C_u_*	10.01	
Curvature coefficient, *C_c_*	1.23	

**Table 4 materials-16-00634-t004:** Main chemical composition of materials used.

	Major oxide (%)	SiO_2_	Al_2_O_3_	CaO	Fe_2_O_3_	MgO	Na_2_O	K_2_O	SO_3_
Materials									
GMTs		59.72	19.06	4.34	5.04	3.78	--	4.44	1.73
FA		49.52	16.68	34.91	0.84	6.67	0.39	0.61	2.14
GFBS		30.85	14.82	41.23	0.63	8.52	2.0	5.7	1.9
CCR		2.83	2.24	68.89	0.25	0.18	0.13	0.21	0.78
MK		52.23	44.01	0.32	0.71	0.31	0.23	0.76	0.38
SH		--	--	--	--	--	99.0	--	--
PG		--	--	41.2	--	--	--	--	58.8

**Table 5 materials-16-00634-t005:** Parameters obtained from the Gaussian curve analysis of solidified/stabilized gold tailings with different confining pressures.

Confining Pressure (kPa)	*s* ^(1)^	*a* ^(2)^	*x_c_* ^(2)^	*w* ^(2)^	*σ* ^(3)^	*R* ^2(4)^
100	1438.79	−15.01	−23.37	22.17	34.43	0.990
200	1645.59	−8.59	−5.52	15.29	30.90	0.991
400	2110.91	−10.68	−1.88	13.43	31.59	0.995

^(1)^ Maximum TCS during curing. ^(2)^
*a* is related to the characteristics and dosages of waste-based binder; *x_c_* and *w* are mainly dominated by confining pressure. ^(3)^ *σ*, standard deviation. ^(4)^ *R*^2^, correlation coefficient.

**Table 6 materials-16-00634-t006:** Strength and deformation parameters of solidified/stabilized gold mine tailings after different curing periods.

Curing Periods (d)	Strength Parameters		Deformation Parameters					
	Cohesive Strength (kPa)	Internal Friction Angle (°)	Secant Modulus (MPa)			Breaking Strain (%)		
			*σ* = 100 kPa	*σ* = 200 kPa	*σ* = 400 kPa	*σ* = 100 kPa	*σ* = 200 kPa	*σ* = 400 kPa
0	142.59	21.16	5.62	9.55	6.73	9.29	8.34	15
3	218.14	20.14	31.47	31.54	35.51	2.24	3.71	3.85
7	205.49	25.18	34.99	40.11	35.24	2.13	2.71	4.23
14	274.65	26.89	70.05	58.85	46.49	1.58	2.25	2.7
28	300.25	31.64	72.61	87.05	60.43	1.31	2.12	3.14
90	323.43	32.18	103.33	133.65	68.97	1.43	1.18	2.89

## Data Availability

Data available on request due to restrictions eg privacy or ethical. The data presented in this study are available on request from the corresponding author.

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
