# Peer review of "Mechanical and Hydration Characteristics of Stabilized Gold Mine Tailings Using a Sustainable Industrial Waste-Based Binder"

_materials, 2023, doi:10.3390/ma16020634_

Round 1

Reviewer 1 Report

Dear Authors,

Please address the following comments and questions:

1. The introduction was not fully developed in relation to the last statements specifically being economical and reliable as well as having great social benefits [Lines 106 to 109]

2. If the formulation cannot be disclosed, how can you readers verify your results? Can you at least provide the basis of your chosen formulation used in this paper?

3. Figure 1 label is not accurate

4. Provide more detail in lines 118 to 130 such as temperature and duration of drying, specifications and operation conditions of ball mill, suppliers of reagents etc.

5. Provide details to determine phase composition of gold mine tailing via XRD

6. Insert XRD diffractogram

7. Data analysis and modelling details including the software used should be included in the methodology

8. Why triaxial compression test was used in this paper?

9. Relate your FTIR and NMR findings to mechanical properties in terms of hydration processes and products

10. Why 3-day and 90-day samples only were subjected to NMR analysis despite termination of hydration was approximated at 40 days? BET results showed S0 S3 and S7 to S90 were somehow grouped and were related to hydration?

11. Why are there discontinuities in Figure 6?

12. Are the equations or models used "units" specific? Please define all variables and parameters accurately

Thank you

Author Response

Authors' Responses to Review Comments

Materials

Title:Mechanical and hydration characteristics of blends gold mine tailings with a sustainable industrial waste-based binder (Manuscript ID: materials-2097171) 

New Title: Mechanical and hydration characteristics of stabilized gold mine tailings using a sustainable industrial waste-based binder

Authors: Zhenkai Pan; Shaohua Hu; Chao Zhang; Guowei Hua; Yuan Li

The authors would like to thank the reviewer's comments, which are very helpful in guiding the authors' rework and revisions of the manuscript.

The responses to the comments are detailed below, in which the paragraphs in regular fonts are the comments by the editor, and the authors' responses are written in italic fonts. In the revised manuscript, the revised sentences are marked in red.

Response to Comments of Reviewer #1

       Please address the following comments and questions:

  1. The introduction was not fully developed in relation to the last statements specifically being economical and reliable as well as having great social benefits.

Response: Thank you very much for this comment. This is a good comment for us. In the introduction part, this paper introduces the potential application of tailings as building materials and raw materials for cement production, and tailings can be also used as a additive for traditional engineering materials. These provide a reference for improving the management of tailings and have economic significance. In addition, it can reduce the harm of tailings discharge to the environment, such as river pollution caused by tailings dam breaking, farmland coverage, which has social benefits. Perhaps the final statement did not match the above well, and look confusing. The authors have deleted the statement that it is economically reliable and has social benefits.

  1. If the formulation cannot be disclosed, how can you readers verify your results? Can you at least provide the basis of your chosen formulation used in this paper?

Response: Thank you very much for this suggestion. After consideration by the various authors, all the authors agree that the formulation will be published, the chemical composition ratios of the binder are shown in Table 1, and the original statement has been changed, as shown in Lines 123-140.

  1. Figure 1 label is not accurate

Response: Thank you very much for this comment. This is a good comment for us. The authors have replaced the original image of the waste-based binder so that Figure 1 in the paper looks clearer.

  1. Provide more detail in lines 118 to 130 such as temperature and duration of drying, specifications and operation conditions of ball mill, suppliers of reagents etc.

Response: Thank you very much for this comment. The authors have re-written this part and supplied more details according to the reviewer's comments. The revised part is shown in Lines 123-140, as follows:

The FA was obtained from a local power plant, of which material with a particle size of less than 45 μm accounted for 80%. The GBFS was collected from a steel plant and then air-dried at 60℃ for 24 h. The  MK was formed by dehydrating kaolin at high temperatures (600~900 ℃) to break the van der Waals bonds between the layered silicate structures. The CCR was acquired by drying an industrial waste liquid from a gas refinery at 105℃ for 8h. The PG, a white powder, was purchased from a by-product company in Hubei province of China. The SH consisting of superior grade pure material with glass bead-like solid particles, was purchased from a chemical products company in Cangzhou city of China. GBFS was mechanically ground in a ball mill (Ransbach-Baumbach, Germany) with a grinding medium of 40 stainless steel balls to increase its specific surface area (SSA).

  1. Provide details to determine phase composition of gold mine tailing via XRD

Response: Thank you very much for this comment. The mineralogy of gold tailings was analyzed by quantitative Rietveld XRD method. The mineralogy of raw tailings mainly contains quartz and illite, and consists of a little bit of clinochlore, pyrite, calcite and galena in decreasing order of content with mineralogical composition component as shown in figure below. The authors have provided more details for determining the composition of the gold tailings, as shown in Lines 160-167.

  1. Insert XRD diffractogram

Response: Thank you very much for this comment. The XRD diffractogram has been supplemented in the introduction section of tailings material.

  1. Data analysis and modelling details including the software used should be included in the methodology

Response: Thank you very much for this comment. The software used in this paper for data analysis and modeling is OriginLab 2018. The modelling is analyzed and fitted by the Gaussian function of the ‘Origin Basic Functions’ in the software-built the nonlinear curve fitting. In section 5.1.2, the methodology and details have been supplemented, as shown in Lines 271-273.

  1. Why triaxial compression test was used in this paper?

Response: Thank you very much for this comment. For uniaxial compression test, it is a convenient and commonly used method to measure the mechanical behavior of materials. However, in addition to mechanical strength, triaxial compression tests can measure the mechanical parameters and observe the failure deformation of stabilized materials.

  1. Relate your FTIR and NMR findings to mechanical properties in terms of hydration processes and products?

Response: Thank you very much for this suggestion. For the result part, the comparison between the result discussion and other literature is relatively weak. The authors are deeply aware of this problem. The authors have rewrited the discussion of the results to make the results of this paper more closely related. The revised part is shown in Lines 445-449 and 453-457.

  1. Why 3-day and 90-day samples only were subjected to NMR analysis despite termination of hydration was approximated at 40 days? BET results showed S0-S3 and S7 to S90 were somehow grouped and were related to hydration?

Response: Thank you very much for this comment. Since the cost of the 27Al MAS-NMR test is relatively expensive, considering that the termination time of the hydration reaction is about 40 days according to model fitting, the NMR analysis was conducted on the samples in the process of hydration and the completion of hydration reaction, that is, the samples are cured for 3 days and 90 days respectively. BET results may reflect that although the hydration reaction was terminated after 40 days of curing, the hydration rate of the sample slowed down after 7 days of curing, so S7 to S90 seems to be a group. 

  1. Why are there discontinuities in Figure 6?

Response: Thank you very much for this comment. The authors thought that the threshold of slow and fast increase in intensity could be more clearly seen in this way, but it did not seem to be. Fig. 6 has been revised, as shown in Lines 318-319

  1. Are the equations or models used "units" specific? Please define all variables and parameters accurately

Response: Thank you very much for this comment. The units of parameters used in all equations and models are specific. The authors have redefined all units of variables and parameters, as shown in Lines 367-370 and 378-381.

Author Response

Authors' Responses to Review Comments

Materials

Title:Mechanical and hydration characteristics of blends gold mine tailings with a sustainable industrial waste-based binder (Manuscript ID: materials-2097171) 

New Title: Mechanical and hydration characteristics of stabilized gold mine tailings using a sustainable industrial waste-based binder

Authors: Zhenkai Pan; Shaohua Hu; Chao Zhang; Guowei Hua; Yuan Li

The authors would like to thank the reviewer's comments, which are very helpful in guiding the authors' rework and revisions of the manuscript.

The responses to the comments are detailed below, in which the paragraphs in regular fonts are the comments by the editor, and the authors' responses are written in italic fonts. In the revised manuscript, the revised sentences are marked in red.

Response to Comments of Reviewer #2

The paper is good in content and writing. Some comments help to improve the paper as follows:

  1. The reason to choose “a content ratio of 5 % binder and 95 % gold tailings” in this study?

Response: Thank you very much for this comment. Although the acquisition of waste-based binder reduces energy consumption and carbon dioxide emission compared with the production of cement, its production also requires a certain cost. Before the study in this paper, the authors conducted a pre-test on the mechanical strength of stabilized/solidified samples containing 3%, 5%, 8% and 10% waste-based binder. While meeting the strength requirements, the cost was reduced to the greatest extent. So a content ratio of 5 % binder and 95% gold tailings was chosen. The reason for selecting the ratio of binder to tailings content has been described more precisely in this paper, as shown in Lines 107-112.

  1. Figure 3 should denote a, b, and c in the figure. Please consider Fig. 3a.

Response: Thank you very much for this comment. This is a good comment for us. The authors have added label in new figure, as shown in Lines 258-261.

Fig. 4. Stress-strain curves and failure deformation of the stabilized/solidified gold tailings: (a) under confining pressure of 100 kPa, (b) under confining pressure of 200 kPa, (c) under confining

pressure of 400 kPa.

  1. How to prepare cylindrical specimens (D=39.1mm, H=80mm) with a controlled dry density of 1.5 g/cm3.

Response: Thank you very much for this comment. The procedures for the solidification/stabilization (S/S) sample preparation was presented in another paper Solidification/stabilization of gold ore tailings powder using sustainable waste-based composite geopolymer’ by the author (attached at the end of the response). If this process is described again in the paper, it will be more complicated, so the original manuscript has been explained in the form of citing references, as shown in Lines 192-193. The cylindrical specimen preparation are as follows:  

          (1) Preparation for mixture. GOTs and WCG are as shown in Fig. 1 (1). The raw materials were weighted in their dry state and thoroughly mixed together in a ball mill. The optimal moisture content was implemented for sample preparation and the materials were mixed to a uniform condition. The targeted water was weighted and added to the mixture. Deionized water was used, with a pH of 6, a temperature of about 22.5 °C and a conductivity of less than 1 µs/m. The obtained mixture prepared for the experiment is shown in Fig. 1 (2).

         (2) Sample moulding. According to the compression state of the mixture, the dry density of the sample is set to 1.5 g/cm3. The correspondingly needed mixture was weighed and added to a sample tub using a funnel, as shown in Fig .1 (3). A block was placed at the bottom of the sample tub. After the prepared mixture was poured into the sample tub, an identical block was placed on top of the sample. These two blocks were squeezed into the sample tub by hammer. Then, the shaped sample was pushed out by a jack. Simultaneously, the blocks were removed, as shown in Figs. 1(4) and (5).

        (3) Sample curing. All the prepared samples were wrapped in a plastic bag to isolate them from air and water, so as to achieve their own hydration reaction. All samples were put in the curing box at a constant temperature (25 °C) for 3, 7, 14, 28, and 90 days before further characterization, as shown in Fig. 1(6).

Fig. 1.  S/S  (Solidification/Stabilization) sample preparation procedures

Pan, Z., Zhang, C., Li, Y., & Yang, C. (2022). Solidification/stabilization of gold ore tailings powder using sustainable waste-based composite geopolymer. Engineering Geology, 309, 106793.

  1. The reason to choose the confining pressure of 100, 200, and 400 kPa.

Response: Thank you very much for this comment. Preliminary test results showed  that the wopt and ρmax of the admixed tailings with 5% of binder dosage were 17.8%, and 1.72 g/cm3, respectively. When the confining pressure is 100, 200, and 400 kPa, the depth of the raw tailings material in the tailings dam is 6, 12, and 24m, respectively. Such confining pressure can not only determine the strength parameters of stabilized/solidified tailings, but also provide a basis for the field application of in-situ grouting construction in tailings dam to improve the stability of tailings dam.

  1. Where is NaOH in this study?

Response: Thank you very much for this comment. The SH consisted of superior grade pure with glass bead-like solid particles, was purchased from a chemical products company in Cangzhou city of China. The original statement has been changed, as shown in Lines 130-132.

  1. In line 348: “Fig. 7 presents the BET curves of waste-based geopolymer solidified/stabilized gold tailings”, what is the term “geopolymer” in this study?

Response: Thank you very much for this comment. This novel waste-based binder belongs to be a geopolymer. ‘Geopolymer’ here are prone to misunderstandings, so the authors have deleted and replaced it, as shown in Lines 382-383.

  1. Please consider the term “geopolymer” in waste-based geopolymer solidified/stabilized gold tailings” as well as in line 269.

Response: Thank you very much for this comment. The “geopolymer” in waste-based geopolymer solidified/stabilized gold tailings of the full manuscript has deleted it and replaced it.

  1. The data presented in Fig. 5, why does the dry density change with curing time?

Response: Thank you very much for this comment. The ettringite and C-S-H gels formed by hydration reaction are calcium-based hydration products with higher density, compared to the raw tailings composition. In general, the higher the density of hydration products, the higher the strength. For example, some papers present relevant researchGarg et al. (2014); Schumacher et al. (2020); Kurda et al. (2018).

Garg, M., & Pundir, A. (2014). Investigation of properties of fluorogypsum-slag composite binders–Hydration, strength and microstructure. Cement and concrete composites, 45, 227-233.

Schumacher, K., Saßmannshausen, N., Pritzel, C., & Trettin, R. (2020). Lightweight aggregate concrete with an open structure and a porous matrix with an improved ratio of compressive strength to dry density. Construction and Building Materials, 264, 120167.

Kurda, R., de Brito, J., & Silvestre, J. D. (2018). Indirect evaluation of the compressive strength of recycled aggregate concrete with high fly ash ratios. Magazine of Concrete Research, 70(4), 204-216.

  1. Figure 9. Please add the value of the vertical axis

Response: Thank you very much for this comment. The ordinate of such figures can not be added value, because if according to the actual value, many curves will overlap in large areas in a figure, which is not conducive to analysis. All curves can only be placed in a coordinate space without ordinate value for analysis such as the figure below, from the literatureZhang et al. (2011); Pan et al. (2022).

Zhang, N., Liu, X., Sun, H., & Li, L. (2011). Pozzolanic behaviour of compound-activated red mud-coal gangue mixture. Cement and concrete research, 41(3), 270-278.

Pan, Z., Zhang, C., Li, Y., & Yang, C. (2022). Solidification/stabilization of gold ore tailings powder using sustainable waste-based composite geopolymer. Engineering Geology, 309, 106793.

  1. The stress-strain presented “in 5.1.1. Stress-strain and deformation characteristics should be clearly presented the type of tests, such as UU, UC, or CD.

Response: Thank you very much for this comment. The curing condition of stabilized/solidified samples in this paper is conventional curing at 20℃ and 90 % humidity, rather than saturated curing environment of immersion in water. If according to the shear test conditions, it should belong to UU. The authors have supplemented the type of tests, as shown in Lines 209-210.

  1. For FTIR results: the peak transmission of 776cm-1 and 522cm-1 is nothing different among curing specimens of 0, 3, 7, 14, 28, and 90 days, please discuss more details.

Response: Thank you very much for this comment. The peak transmission of 776cm-1is associated symmetric Si―O stretching vibration. This band corresponds to quartz. The quartz did not participate in the reaction during the hydration process, so all stabilized samples had similar peak transmission. The authors have revised and discussed more details, as shown in Lines 445-449.

  1. The abstract and conclusion should be written with more specific content.

Response: Thank you very much for this comment. This is good comment for us. In order to the abstract and conclusion be more specific, the abstract and conclusion have been re-writed, as shown in Lines 15-35 and 518-524. The findings and content showed in the revised abstract and conclusion were more clear.

Reviewer 3 Report

-        The paper is discussing the Mechanical and hydration characteristics of blends gold mine 2 tailings with a sustainable industrial waste-based binder. Many things have been studied. But, it seems that none has been studied thoroughly. Some explanations are unconvincing. There are some problems to be addressed before publication.

-        The findings in the abstract are not clear, authors should re-write the abstract.

-        English language of abstract is very weak.

-        Introduction of this study is very weak from both sides, language and academics.

-        The author might consider citing these papers: 1. "Effect of nano-silica on the chemical durability and mechanical performance of fly ash based geopolymer concrete", 2. "Mechanical and durability properties of fly ash and slag based geopolymer concrete".

-        The section on materials and specimen preparation needs to rephrase.

-        Figure 1 is not clear; the author might improve it.

-        In the results section, the discussion of the results is mostly weak and not compared to other studies. Therefore, the section needs to rewrite.

-        In the results section, the quality of all the figures is poor.

-        Conclusions: The discussion about technological benefits has to be separated in the article according to the points of conclusions. The analysis of the results is quite basic and deserves better and deeper processing.

-        Regarding the above comments we found this manuscript includes interesting outcomes and can be published after the above revisions. Therefore, it needs major revisions.

Author Response

Authors' Responses to Review Comments

Materials

Title:Mechanical and hydration characteristics of blends gold mine tailings with a sustainable industrial waste-based binder (Manuscript ID: materials-2097171) 

New Title: Mechanical and hydration characteristics of stabilized gold mine tailings using a sustainable industrial waste-based binder

Authors: Zhenkai Pan; Shaohua Hu; Chao Zhang; Guowei Hua; Yuan Li

The authors would like to thank the reviewer's comments, which are very helpful in guiding the authors' rework and revisions of the manuscript.

The responses to the comments are detailed below, in which the paragraphs in regular fonts are the comments by the editor, and the authors' responses are written in italic fonts. In the revised manuscript, the revised sentences are marked in red.

Response to Comments of Reviewer #3

The paper is discussing the Mechanical and hydration characteristics of blends gold mine tailings with a sustainable industrial waste-based binder. Many things have been studied. But, it seems that none has been studied thoroughly. Some explanations are unconvincing. There are some problems to be addressed before publication.

  1. The findings in the abstract are not clear, authors should re-write the abstract. English language of abstract is very weak. Introduction of this study is very weak from both sides, language and academics. The author might consider citing these papers: 1. "Effect of nano-silica on the chemical durability and mechanical performance of fly ash based geopolymer concrete", 2. "Mechanical and durability properties of fly ash and slag based geopolymer concrete".

Response: Thank you very much for this suggestion. The abstract has been re-writed, as shown in Lines 15-35. The findings showed in the revised abstract were more clear, and English writing was proofreaded by a English-speaking professor.

Introduction of this study is not smooth and deep enough. The authors have quoted and referred to these two articles according to the opinions of the reviewers, which is really beneficial for Introduction, as shown in Lines 85-88 and 92-95.

  1. The section on materials and specimen preparation needs to rephrase.

Response: Thank you very much for this suggestion. For section on materials and sample preparation, the authors have re-described this part,  as shown in Lines 123-140 and 160-167.

  1. Figure 1 is not clear; the author might improve it.

Response: Thank you very much for this suggestion. This is a good comment for us. The authors have replaced the original image of the waste-based geopolymer binder so that Figure 1 in the paper looks clearer, as shown in Lines 141-142.

  1. In the results section, the discussion of the results is mostly weak and not compared to other studies. Therefore, the section needs to rewrite. In the results section, the quality of all the figures is poor.

Response: Thank you very much for this suggestion. For the result part, the comparison between the result discussion and other literature is relatively weak. The authors are deeply aware of this problem. The authors have rewrited the discussion of the results and supplemented the references as follows. The following references make the results of this paper more reliable and more closely related to the concatenation of various test results, as shown in Lines 445-449 and 453-456.

GülÅŸan, M. E., Alzeebaree, R., Rasheed, A. A., NiÅŸ, A., & KurtoÄŸlu, A. E. (2019). Development of fly ash/slag based self-compacting geopolymer concrete using nano-silica and steel fiber. Construction and Building Materials, 211, 271-283.

Alzeebaree, R., Çevik, A., Nematollahi, B., Sanjayan, J., Mohammedameen, A., & GülÅŸan, M. E. (2019). Mechanical properties and durability of unconfined and confined geopolymer concrete with fiber reinforced polymers exposed to sulfuric acid. Construction and Building Materials, 215, 1015-1032.

  1. Conclusions: The discussion about technological benefits has to be separated in the article according to the points of conclusions. The analysis of the results is quite basic and deserves better and deeper processing.

Response: Thank you very much for this suggestion. In view of the technical benefits of this paper, this paper re-considers to publish the proportion of binder formula in section on materials, so that readers can better verify our results, and the analysis of the results for more in-depth processing, as shown in Lines 123-125 and 518-524.

Round 2

Reviewer 1 Report

Thank you for your revisions. 

Proof read your manuscript.

Format your tables such as Table 2, Table 4 etc as prescribed by the publisher.

Author Response

Format your tables such as Table 2, Table 4 etc as prescribed by the publisher.

Response: Thank you very much for this comment. The author did not find the format of the journal 's prescribed table, which would be formatted as prompted by the assistant editor.